# A Novel Approach for Emotion Detection and Sentiment Analysis for Low Resource Urdu Language Based on CNN-LSTM

Farhat Ullah [1], Xin Chen [1], Syed Bilal Hussain Shah [2,*], Saoucene Mahfoudh [2], Muhammad Abul Hassan [3,*] and Nagham Saeed [4]

[1] School of Automation, China University of Geosciences, Wuhan 430074, China
[2] School of Computing and Informatics, Dar Al-Hekma University, Jeddah 22246, Saudi Arabia
[3] Department of Information Engineering and Computer Science, University of Trento, 38122 Trento, Italy
[4] School of Computing and Engineering, University of West London, London W5 5RF, UK
* Correspondence: sshah@dah.edu.sa (S.B.H.S.); muhammadabul.hassan@unitn.it (M.A.H.); Tel.: +92-3414-340624 (M.A.H.)

**Abstract:** Emotion detection (ED) and sentiment analysis (SA) play a vital role in identifying an individual's level of interest in any given field. Humans use facial expressions, voice pitch, gestures, and words to convey their emotions. Emotion detection and sentiment analysis in English and Chinese have received much attention in the last decade. Still, poor-resource languages such as Urdu have been mostly disregarded, which is the primary focus of this research. Roman Urdu should also be investigated like other languages because social media platforms are frequently used for communication. Roman Urdu faces a significant challenge in the absence of corpus for emotion detection and sentiment analysis because linguistic resources are vital for natural language processing. In this study, we create a corpus of 1021 sentences for emotion detection and 20,251 sentences for sentiment analysis, both obtained from various areas, and annotate it with the aid of human annotators from six and three classes, respectively. In order to train large-scale unlabeled data, the bag-of-word, term frequency-inverse document frequency, and Skip-gram models are employed, and the learned word vector is then fed into the CNN-LSTM model. In addition to our proposed approach, we also use other fundamental algorithms, including a convolutional neural network, long short-term memory, artificial neural networks, and recurrent neural networks for comparison. The result indicates that the CNN-LSTM proposed method paired with Word2Vec is more effective than other approaches regarding emotion detection and evaluating sentiment analysis in Roman Urdu. Furthermore, we compare our based model with some previous work. Both emotion detection and sentiment analysis have seen significant improvements, jumping from an accuracy of 85% to 95% and from 89% to 93.3%, respectively.

**Keywords:** emotion detection; sentiment analysis; Roman Urdu; machine learning; deep learning

## 1. Introduction

Nowadays, Electronic Data Extraction Techniques (EDET) have become commonplace, extending from instant messaging apps to automated archives with millions of data. The increase in data has given rise to numerous new difficulties. However, one effort is being made to automatically classify some of this textual information to make it easier for users to retrieve, analyze, and edit data to generate patterns and knowledge. Many individuals and businesses are becoming interested in organizing enormous amounts of electronic data into categories. Text classification (TC) is the only way to solve this challenge. TC is the process of organizing text into predetermined groups. It makes use of a wide variety of expertise, such as artificial intelligence (AI), natural language processing (NLP), machine learning (ML), and deep learning (DL). This technique relies on supervised learning (SL), in which

we "train" a model by feeding it a massive quantity of information. TC has a wide range of applications, including emotion recognition (ER), topic modelling (TP), sentiment analysis (SA), intent detection (ID), and spam detection (SD). ED strongly influences the extent to which an individual is invested in a field, and this is especially true in contexts where humans use nonverbal cues such as gestures [1], facial expressions [2], voice pitch [3], and the choice of words to explain their emotions [4]. SA offers an overview of public opinions expressed by people through blogs and reviews about a specific event, object, activity, topic, product, and service, which may be a huge boon to making the right choices. The terms SA and opinion mining (OM) are increasingly being used interchangeably to refer to the same topic of study, namely the identification of polarity and emotion in online discussions. In general, emotion is defined as a powerful sensation, such as compassion, passion, anxiety, joy, or sorrow, whereas sentiment is the general polarity.

Emotion and SA classifications using English and other different language text data with abundant resources have received considerable attention. However, because there are not enough corpora with labels, emotion and SA categorization have not been thoroughly explored in languages with limited resources, such as Urdu. More than 100 million individuals worldwide consider Urdu their first language. Urdu, the language spoken officially in Pakistan, is of the Indo-Aryan family. In addition, Urdu is widely spoken in both India and Bangladesh. They do it by posting emotional messages in Urdu across a variety of social media sites [5,6]. Because of Urdu's complicated morphology, ED and SA are more challenging than English. Urdu is difficult to understand due to a variety of issues. Its free word order, rich morphology, and multidimensional spelling are among them. This makes the job of Urdu SA and ED even more difficult. Due to the reasons above, ED and SA methods created for other languages are not necessarily applicable to Urdu. Figure 1 shows a translation from Urdu to Roman Urdu and English.

**Figure 1.** Translation from Urdu to Roman Urdu and English.

Although text can transmit a wide range of emotions, psychologists have focused on the most fundamental of these categories. Keltner et al. [7] investigated the six primary emotions of love, joy, fear, anger, disgust, and surprise, while Plutchik [8] proposed a list of eight primary emotions that included aspiration and trust. Our study focuses on five emotions: fear, happiness, anger, sadness, and love. If an instance lacks these emotions, we label it neutral. Many methods for classifying emotions have been developed by researchers, incorporating lexicon-based techniques (LBTs). This example shows how LBTs struggle to handle the complex nature of opinions. To deal with the complexity and failure of lexicon-based approaches, our main contributions are as follows:

- Labelled sentences for ED and SA were collected from the different sites, with additional labels added manually.
- Finally, the five most common emotions in Urdu, which we call Urdu emotion detection (UED), are the focus of our contribution (happiness, sadness, fear, anger, and love). Furthermore, for sentiment analysis, our contribution is to focus on positive and negative sentences.
- Quality control of a corpus can be achieved by using an inter-annotator agreement to validate the accuracy of the annotated data.

- A DL-based model for identifying ED and SA in corpora is proposed, as well as a word-embedding method for the corpus.

The remainder of the paper is organized as follows: The literature on the various ED and SA corpora and the various methodologies used to categorize them, will be discussed in Section 2. Section 3 discusses the experimental setting, whereas Section 4 presents our findings and analyses from multiple perspectives. Finally, Section 5 brings our efforts to a close and outlines our plans for the future.

## 2. Literature Review

In this section, we will briefly discuss sentiment and emotion analysis and then discuss some related work. As a development of sentiment analysis, emotion analysis is described as a linguistic procedure for recognizing emotions expressed in written material. According to Yadollahi et al. [9], sentiment and emotion have a strong association in which one can form an opinion about anything based on their feelings, and the opposite is also true. Numerous emotions exist, each one corresponding to a distinct range of human experiences. These distinct feelings have an impact on how people act. The ED corpus is a multilingual database of textual information encompassing both English and Urdu, and its contributions have been explored in the literature. Table 1 provides seven sections that summarize previous research on emotion recognition: author name, Models Applied, Purpose, Contribution, Result, Language, and Limitation.

**Table 1.** Emotion Detection Previous Literature Review.

| Author Name | Models Applied | Purpose | Contribution | Accuracy | Language | Limitations |
|---|---|---|---|---|---|---|
| Rabail et al. [10] | NB, SVC, LR, RF, KNN, SVCL | Features classification | Provide dataset for other researchers | 55% | Roman Urdu | Low accuracy |
| Jonathan Herzig et al. [11] | SVM | Features classification | Low computational power | 62% | English | Used small data set and achieved less accuracy |
| Adil Majeed et al. [12] | KNN, DT, RF, SVM | Features classification | Compared different classifier models | 69.4%. | Roman Urdu | Achieved 69.4%. accuracy |
| Raza Ali [13] | LSTM, CNN, LR, RF, SVM | Features classification | Improved accuracy | 73% | Roman Urdu | The result can be improved |
| Kevin et al. [14] | Survey | To introduce NLP to social researchers | Demonstrate the limitations of NLP tools | 75%. | English | Not deeply demonstrated |
| Yves Bestgen [15] | SVM | Features classification | Correctly classified emotion | 78% | English | The achieved accuracy is low |
| Sara Durrani [16] | SAVEE, EMOO, IEMOCAP | Features classification | Improved accuracy | 80.34% | English | Need to check in other low languages |
| Maryam et al. [17] | NB, SVM DT | Features classification | Comparative analysis | 90% | English | Still, accuracy needs to improve |
| Noman Ashraf et al. [18] | CNN, LSTM | Features classification | Improved accuracy to some extent | 90% | Urdu | A not good representation of corpus |
| Muhammad et al. [19] | LSTM, MLP, BI-LSTM | Features classification | Improved accuracy | 90.2% | Urdu | Still, accuracy needs to improve |
| Abdul et al. [20] | SMO, DT, SVM | Classification | Improved accuracy | 92% | English | Need to extract more sensitive information |
| Taimoor et al. [21] | LSTM, CNN | Classification | Take the initiative for hierarchal data | 94% | Urdu | Time complexity |
| Usama Khalid et al. [22] | Bilingual models | Classification | Used to defeat multilingual modeling | 95% | English, Roman Urdu | Face complexity in the downstream task |

Sentiment analysis (SA) generates an overview of the public thoughts expressed by people about a specific activity. SA combines NLP and Data Mining techniques which significantly aid in effective decision-making. SA and other developing fields are often used interchangeably to detect polarity and emotions. Sentiment analysis is a three-level classification procedure. These are the sentence level, the aspect level, and the document level. Using sentence-level sentiment analysis, each sentence is assigned a positive, negative, or neutral sentiment label. Aspect-level sentiment analysis is accomplished by assigning positive or negative evaluations to the entities' features. At the document level, the entire document is regarded as a single topical unit of basic information. The document is termed as positive if there are more positive sentences than negative sentences; it is termed as negative if there are more negative sentences than positive ones. Table 2 shows the previous literature on sentiment analysis.

**Table 2.** Sentiment Analysis Previous Literature Review.

| Author Name | Models Applied | Purpose | Contribution | Accuracy | Language | Limitations |
|---|---|---|---|---|---|---|
| Faiza et al. [23] | SVM | Feature Classification | Classified eCommerce site | 60% | Roman Urdu | Need to improve accuracy |
| Neelam et al. [24] | SVM, KNN | Feature Classification | Improved accuracy | 80.6% | English | Used small dataset |
| Yang et al. [25] | Survey | Classification | Improved accuracy | 82.5% | English | Need to improve accuracy |
| Md et al. [26] | K-NN, NB, DT | Classification | Suggested social media dataset | 90% | English | Used small dataset |
| Hussain et al. [27] | RNN, LSTM. | Model comparison | An efficient model for SA | 92% | Roman Urdu | Used small dataset |
| Huniya et al. [28] | SVM, kNN, DT, etc. | Classification | Compared classification models | 93% | Roman Urdu/Hindi | Need features selection with large data |
| Nazish et al. [29] | SLR | Compare different models | Explored previous studies | N/A | Urdu | Further analysis is needed |
| L.khan et al. [30] | CNN-LSTM | Classification | Improved accuracy | 90% | English/ Roman Urdu | More research is required to improve the outcome. |
| C.Octavian et al. [31] | CNN-BiLSTM, LSTM, BiGRU | Classification | Improved accuracy | 75.60 | Contextual Cues | Other embeddings methods with the same model must be tested. |

Early, accurate and immediate diagnosis of malaria detection and prediction using conventional methods are deemed ineffective because of their approaches, which include the necessity of time-intensive and poor performance. Moreover, limitations of previous studies using ML and DL are related to the small size of the datasets used and the limited number of features without the process of feature selection. It has been observed that the provided dataset has a high imbalance between emotions detection and sentiment analysis cases.

## 3. Proposed Methodology

The following content provides an overview of the proposed system for data annotation with each part of the dataset framework explained in Figure 2. The suggested model collects raw data from various sources and cleans it up in a pre-processing stage to eliminate

duplicates, URL links, new lines, and consistency in spelling. Annotators provide the data for annotation after it has been pre-processed, in which emotion labels are assigned to the data. In addition, we use count vectorization to extract features from the data, and then we split the dataset into a training set and a testing set. After feature extraction, the deep learning models convolution neural network (CNN) are trained with LSTM on training data and then tested. The Adam optimizer assesses the model and fine-tunes it for the best performance.

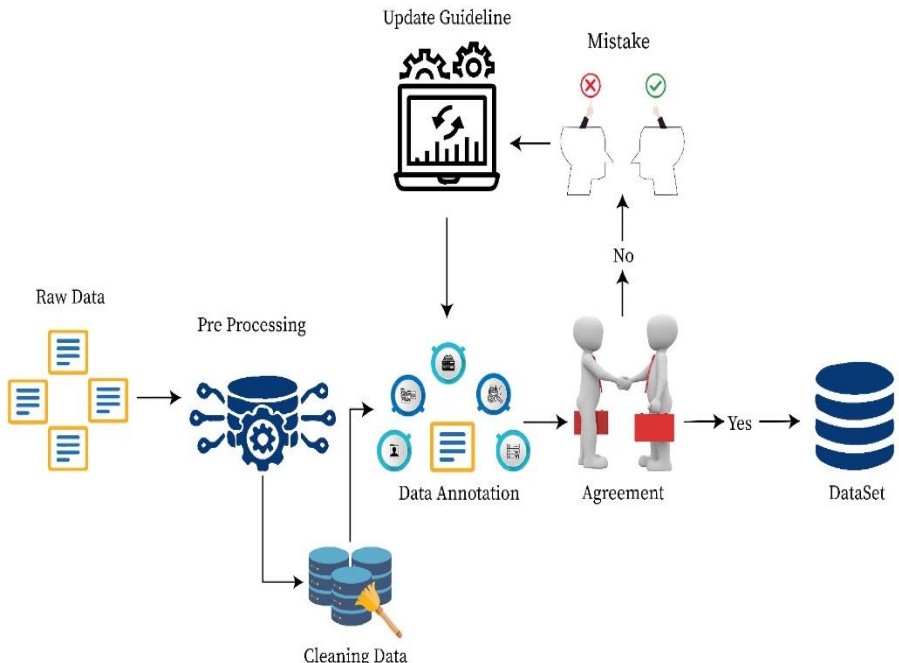

**Figure 2.** Explain Data Annotation Technique.

### 3.1. Pre-Processing

A raw corpus from multiple web sources has space limits, erroneous characters, improper formats, noise errors, etc. So, we cleaned and changed the data into the proper format before applying a classification algorithm to the processed corpus. Figure 3 depicts the proposed methodology processing steps, with the following details below:

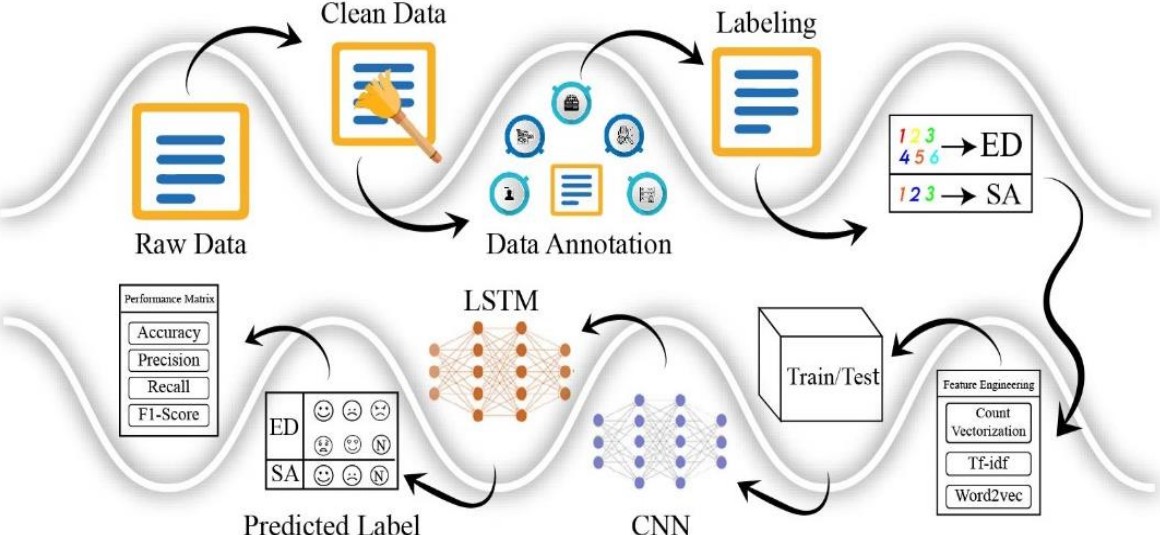

**Figure 3.** Proposed Methodology for Emotion Detection and Sentiment Analysis.

### 3.1.1. Noise Removal

The results of an emotion analysis task may be impacted by the presence of non-emotional characters such as the @ symbol, the # sign, microblog expressions (ME), HTML tags, URL links, new lines, and so on. A deletion or replacement strategy can be employed for these data. Regular expression erases all characters except microblog expressions, which are substituted with Roman Urdu text (RUT).

### 3.1.2. Spelling Uniformity

There are various spelling variations for the same term because the basic spelling guidelines for Roman Urdu are not specified. We randomly picked a corpus and manually labelled sentences with multiple versions to measure the diversity in pronunciation in Roman Urdu. All sentences are measured using these terms.

### 3.1.3. Space Problem

Space insertion and deletion difficulties are also common in Roman Urdu writing. When the text is entered casually, a word is wrongly split into two different words, although it is the correct term.

### *3.2. Data Annotation*

In this phase, we want to automate Roman Urdu text with emotion detection and sentiment analysis. An emotion-labelled corpus was required for this task. Table 3 displays the six primary emotions that make up the emotional spectrum. However, as seen in Table 4, three distinct classes make up what is known as sentiment analysis. As is normal practice, the annotated data sets are generated by trained professionals. These were the rules that we requested the annotators to follow for emotion detection.

1. Assign a class to a given sentence from one of the six categories of emotion described above.
2. If the given sentence has no instances of the emotion class, it should be marked as neutral.
3. If a sentence is linked to several emotions, then the context determines which emotions are closest.

**Table 3.** Selected Emotion Examples with Label.

| S.NO | Emotion | Relation | Label |
|------|---------|----------|-------|
| 1. | Happy | Feeling or showing pleasure or contentment | Assigning label 1 |
| 2. | Sad | Feeling or showing sorrow; unhappy | Assigning label 2 |
| 3. | Anger | Something you feel has deliberately done you wrong | Assigning label 3 |
| 4. | Love | Characterized by intimacy, passion, and commitment | Assigning label 4 |
| 5. | Fear | Natural, powerful, and primitive human emotion | Assigning label 5 |
| 6. | Neutral | Other categories | Assigning label 6 |

For sentiment analysis, we request annotators to follow these rules:

1. Assign a class to a given sentence from one of the three categories of sentiment described above.
2. If the given sentence has no instances in the positive or negative class, it should be annotated as neutral.
3. The context determines the closest feelings if an example belongs to both defined classes.

**Table 4.** Sentiment Analysis Examples with Label.

| S.NO | Emotion | Relation | Label |
|------|---------|----------|-------|
| 1. | Positive | Happy or excited | Positive |
| 2. | Negative | Angry, annoyed, or frustrated | Negative |
| 3. | Natural | Seem satisfied but do not express | Assigning label 3 |

We used keyword-based data generation for emotion detection and sentiment analysis to uncover other emotion-related sentences. To begin, we discovered many keywords for each emotion detection, as shown in Table 5. Then, depending on those keywords, we extracted sentences and sent them to annotators.

**Table 5.** Keywords Sample for Emotion Detection.

| Emotion | HAPPY | | | SAD | | |
|---------|-------|---------|------|-----|---------|------|
| | **Roman Urdu** | **English** | **Urdu** | **Roman Urdu** | **English** | **Urdu** |
| 1 | Shandar | Glorious | شاندار | Moat | Death | موت |
| 2 | Faida | Benefit | فائدہ | Preshan | Anxiety | بے چینی |
| 3 | itmad | Confidence | اعتماد | Dhamakha | Blast | دھماکہ |
| 4 | Behtreen | Excellent | بہترین | udasi | Sorrow | اداسی |
| 5 | Mazak | Joke | مذاق | Bimar | Sick | بیمار |
| 6 | pur kashish | Attractive | پر کشش | khudkashi | Suicide | خودکشی |
| 7 | fakhr | Pride | فخر | nuqsan | Loss | نقصان |
| 8 | inam | Prize | انعام | burai | Evil | برائی |
| **Emotion** | **Fear** | | | **Anger** | | |
| 1 | Khof Zadah | Scared | ڈرا ہوا | Jarhana | Aggressive | جارحانہ |
| 2 | Khoofnaq | Dreadful | خوفناک | tashadud | Violence | تشدد |
| 3 | Bitarteeb | Unsettled | بے ترتب | Antiqame | Retaliation | انتقامی |
| 4 | Beitimadi | Distrustful | بے اعتمادی | Ghussa | Rage | غصہ |
| 5 | Kanpna | Shudder | کانپنا | Larna | Fight | لڑنا |
| 6 | Darpok | Timid | ڈرپوک | Dushmani | Hostility | دشمنی |
| 7 | Ghermahfooz | Insecure | غیر محفوظ | thaparmarna | Slapped | تھپڑ مارا |
| 8 | Kheran | Shocked | حیران | Nafratangez | Hateful | نفرت انگیز |
| **Emotion** | **Neutral** | | | **Love** | | |
| 1 | Muft | Free | مفت | Muhabat | Love | محبت |
| 2 | assan | Easy | آسان | tarif karen | Admire | تعریف کریں |
| 3 | heran | Amazed | حیران | jazbah | Passion | جذبہ |
| 4 | praatmad | Confident | پر اعتماد | hamdardi | Sympathy | ہمدردی |
| 5 | qasam | Kind | قسم | romania | Romantic | رومانوی |
| 6 | Qabool | Receptive | قبول | metha | Sweet | میٹھا |
| 7 | Dilchaspi | Interested | دلچسپی | mukhlis | Sincere | مخلص |
| 8 | samaj | Understanding | سمجھ | khoobsurat | Pretty | خوبصورت |

The same as emotion detection, we also discovered a large number of keywords for sentiment analysis, as shown in Table 6. Then, depending on those keywords, we extracted sentences and sent them to annotators.

**Table 6.** Positive and Negative Keywords Sample for Sentiment Analysis.

| Emotion | Positive | | | Negative | | |
|---------|----------|---------|------|----------|---------|------|
| | Roman Urdu | English | Urdu | Roman Urdu | English | Urdu |
| 1 | Itmeenan | Satisfaction | اطمنان | Darna | Afraid | ڈرنا |
| 2 | Benafsi | Altruism | بي نفسی | alarm | Alarm | الارم |
| 3 | Madad | Relief | مدد | Naraz | Annoyed | ناراض |
| 4 | Piyar | Affection | پیار | bor | Bored | بور |
| 5 | Khush mazaji | Cheerfulness | خوش مزاجی | Talah | Bitter | تلخ |
| 6 | Tareef | Admiration | تعرف | Zabardasti | Coercive | زبردستی |
| 7 | Josh | Euphoria | جوش | Aljao | Confusion | الجهاؤ |
| 8 | Qinat | Contentment | قناعت | Tauheen | Contempt | توہین |

### 3.3. Validation of Corpus

A significant portion of this work was carried out using the corpus validation approach. Eliminating misleading or unclear records for sentiment analysis and emotion detection is the main motivation. In many cases of outliers, the training set could be misleading if such entries were left in. During the annotation phase, the annotator is given each input randomly and asked to label it with one of six possible emotional states. Additionally, when the annotator cannot identify the appropriate emotional class, an additional condition is imposed for an unclear situation. Each sentence is reviewed by many different annotators, as explained in Equations (1) and (2). Assuming an emotion label is valid if a group of annotators all uses the same label. At the conclusion of annotations, if not all group members vote in favour, the corresponding entry is eliminated from the corpus.

$$V_{Data} = f\left(\Sigma_{1+1}^n < \lim\right) \tag{1}$$

$$V_D = \sum_{Data}^n Anno_\forall \text{ agree (or) } \sum_{Data}^n Anno_\forall \neq \text{agree} \tag{2}$$

### 3.4. Feature Extraction

Three methods are used to extract features: term frequency and inverse document frequency (Tf-idf), count vectorization, and word embeddings. Count vectorization uses our vocabulary dimensions to create vectors. When a word is used in a sentence, it is assigned a dimension. Each time that word is used, one is added. Tf-idf deals with how often a word appears in a sentence and how significant its occurrences are. Tf-idf is calculated using two measures: a word's frequency in the sentence and its inverse frequency with other words shown in Equations (2)–(13). Computers are unable to read and comprehend textual information created by humans. Furthermore, we use a word embedding technique to translate the Urdu textual data into a numerical vector form suitable for a machine's data processing.

Explanation: Information theory can be used to develop term frequency and inverse document frequency (tf-idf). Understanding why their product has value in terms of the combined informative content of a document is helpful. Equation (3) is a defining assumption regarding the distribution p (d, t). In the case where a document in the data D contains a specific phrase, t, the conditional entropy of the document is Equations (4)–(7).

In addition, in the context of the notation, D and T are referred to as "random variables," where D stands for "draw a document" and T stands for "term." The knowledge shared between parties can be represented by Equations (8)–(10). The last thing that needs to be done is to enlarge Pt, which stands for the unconditional probability of drawing a term, regarding the arbitrary selection of a document in Equations (11)–(13).

$$p(d|t) = \frac{1}{|\{d \in D : t \in d\}} \tag{3}$$

$$H(D|T = t) = -\sum_d Pd|t \, Log \, P \, d|t \tag{4}$$

$$-Log\frac{1}{|\{d \in D : t \in d\}|} \tag{5}$$

$$-Log\frac{|\{d \in D : t \in d\}|}{|D|} + \log |D| \tag{6}$$

$$-idf(t) + \log |D| \tag{7}$$

$$M(T; D) = H(D) - H(D|T) \tag{8}$$

$$\sum_t Pt \, . \, (H(D) - H(D|W = t)) \tag{9}$$

$$\sum_t Pt \, . \, idf(t) \tag{10}$$

$$M(T; D) = \sum_{t,d} Pt|d \, . \, pd.idf(t) \tag{11}$$

$$M(T; D) = \sum_{t,d} tf(t,d). \frac{1}{D}. \, idf(t) \tag{12}$$

$$= \frac{1}{D} \sum_{t,d} tf(t,d). \, idf(t) \tag{13}$$

Google's Word2Vec is a popular word vector tool based on CBOW and skip-gram models. Word vectors are trained using the skip-gram model (SGM) on substantial unmarked data, and these vectors are then used as inputs to the proposed model. Two primary methodologies can be used to build and deploy a likelihood and negative sampling in the skip-gram neural network model (NEG). The suggested model uses the likelihood approach, and its mathematical representation can be found in [14–25].

The primary concept that underlies the algorithm is that to begin, we initially choose a random value for each word in the vocabulary's vector. After placing all the context words at position t, we use the Equation (14) to determine which context words are most likely to occur given the center word. To simplify the process of finding derivatives and turning this equation into a minimisation problem, all we need to do is take the log of the equation and multiply it by −1. This will allow us to calculate the negative log-likelihood in Equation (15). Since we have included the logarithm in the equation, it is time to switch from multiplication to addition in Equation (16).

$$L(\theta) = \prod_{T=1}^{T} \prod_{-m \le j \le m} P(Wt + j|Wt; \, \theta) \tag{14}$$

$$j(\theta) = -\frac{1}{T}\log L(\theta) \tag{15}$$

$$-\frac{1}{T}\sum_{t=1}^{T} \sum_{-m \le j \le m} \log P(Wt + j|Wt; \, \theta) \tag{16}$$

Furthermore, we estimate the likelihood of the context word given the primary term in Equation (17). This will be achieved by employing two sets of vectors, referred to as Uw and Vw, to represent each word. Uw is used when w is part of the context; Vw is used when w is the main idea of the phrase. The following form will emerge from including these two vectors in our probability computation for the context word c and the center word o.

$$P(O = o | C = c) = \frac{\exp(U_o^T V_c)}{\Sigma_{w \epsilon \text{ Vocab}} \exp(U_w^T V_c)} \tag{17}$$

We will use the gradient descent technique in Equations (18) and (19) to gradually change all the weights to reach the highest likelihood. We will be able to identify the path we need to take in order to update the weights by taking into account the derivative of our loss function concerning both U and V.

$$\frac{\partial}{\partial v_c} \exp(\log \exp(\exp(U_o^T V_c))) \tag{18}$$

$$\frac{\partial}{\partial v_c} \exp(U_w^T V_c) = U_0 \tag{19}$$

The following equation results from moving the derivative of log(x) inside the summation. Take the derivative of log(x) to proceed to the second half of the problem (20). To take the derivative of the term exp(x) and change the sign of the sum, add Equation (21). If we carefully study Equations (20) and (21), we will see that the term in the summation is the same as the probability term we previously discussed. After obtaining Equation (22), we will incorporate these data in Equation (23).

$$\frac{1}{\sum_{w=1}^v \exp(U_w^T V_c)} \sum_{x=1}^v \frac{\partial}{\partial v_c} \left( U_x^T V_c \right) \tag{20}$$

$$\sum_{x=1}^v \frac{\exp\left(U_x^T V_c\right)}{\sum_{w=1}^v \exp(U_w^T V_c)} * U_x \tag{21}$$

$$\sum_{x=1}^v P(x|c) * U_x \tag{22}$$

$$\frac{\partial j(\theta)}{\partial v_c} = -u_o + \sum_{x=1}^v P(x|c) * u_x \tag{23}$$

Using the same methodology, we can also determine the derivative of J(θ) to Uw. Uw will have two distinct applications: one for situations in which the word w is present in the context and another for situations in which it is absent. In either case, the derivatives' output will equal Equations (24) and (25).

$$w \neq O \quad \frac{\partial j(\theta)}{\partial u_w} = \sum_{x=1}^v P(x|c) * V_c \tag{24}$$

$$w = O \quad \frac{\partial j(\theta)}{\partial u_w} = V_c + \sum_{x=1}^v P(x|c) * V_c \tag{25}$$

### 3.5. Convolution Neural Network (CNN)

In order to predict the emotion and sentiment analysis class based on text input, our model is based on two feature learners, CNN and LSTM. CNN is a popular ANN architecture used for image classification and object detection. However, it also works well in NLP, and recommender systems explained in Algorithm 1, which illustrates the full proposed algorithm for ED and SA from Urdu texts.

---

**Algorithm 1:** Proposed Methodology

---

Input: Dataset Frames (DF)
**Output:** Emotion and Sentiment Analysis
**Begin**
**Procedure** Data_Gen (Multiple sources)
data ←scrap_data (Sources)
**if** (dataset ≠ empty)
    **if** (data == space)
        **then** text ←split(data)
    **if** ((text == limit-length
        **then** sentence ←text
    **Else**
        drop(text)
        **Return**
**Procedure** process annotation ()
    url ←website_url
    ED or SA ←labels Assign
    **Return** sentence
**Procedure** (Df)using in Sequence CNN
    **for** each T ∈ sentence **do**
        Df ←Nt (sentence)
        Ut ← DR(Df)
        ED or SA ← CNN (word2vec or Tfidf or CV(T))
        **Return** Feature
**Procedure** classification LSTM
    **for** each T ∈ sentence do
        Feature ← Classification
        **Return** Emotion
**End**

---

The main reason for using CNN is that it automatically extracts useful features from input data, as shown in Equations (26) and (27). We used 300 numbers filters and stride 1 for the CNN layer and then apply global average pooling in the pooling layer, which is more significant and understandable. By using more robust local modelling, it ensures that feature maps correspond to their respective categories. Furthermore, we pass the resultant feature to the LSTM classifier.

$$x_j^i = f(\Sigma_{i \in Mj} x_i^{l \cdot 1} * k_{ij}^l + b_j^l) \tag{26}$$

$$x_j^i = f\left(B_j^l \text{ down} \left(x_j^{l \cdot 1}\right) + b_j^l\right) \tag{27}$$

### 3.6. Long Short-Term Memory

Long short-term memory (LSTM) is an emerging variation of the RNN model that is frequently used to overcome overflow or vanishing error gradients and capture long-term dependencies. By controlling the error gradient with its gates, LSTM can get around this issue. The LSTM is represented mathematically in Equations (28)–(34) as follows:

$$h_t = f(W * t + Uht - 1 + b) \tag{28}$$

$$I_t = \sigma(W_i * t + U_i ht - 1 + b_i) \tag{29}$$

$$f_t = \sigma(W_f * t + U_f ht - 1 + b_f) \tag{30}$$

$$O_t = \sigma(W_o * t + U_o ht - 1 + b_o) \tag{31}$$

$$g_t = \sigma\left(W_g * t + U_g ht - 1 + b_g\right) \tag{32}$$

$$c_t = (f_t \ominus c_t - 1 + i_t \ominus g_t) \tag{33}$$

$$h_t = (o_t \Theta \tan h(c_t)) \tag{34}$$

The overall result of the LSTM component, which serves as the input for SOFTMAX, is taken to be h64. When the signal has gone through the SOFTMAX, the category decision is given as a probability, as shown in Equation (35).

$$S(c_i) = \frac{\exp(c_i)}{\Sigma_j \exp(c_j)} \tag{35}$$

### 3.7. Optimization Algorithm

Successful implementation of a deep neural network model depends on the optimization algorithm [32–35]. Accurate global optimal solutions can be found using an effective optimization method in a short time, and connected neurons' weight matrixes can be reliably updated. The optimization algorithm's main objective is to discover the global optimal solution via gradient descent in neuron backpropagation and update neuron connection weight; these two tasks constitute the bulk of the algorithm's processing time. Adadelta, RMSprop, Adam, Stochastic Gradient Descent (SGD), Adamax, etc., are all examples of popular optimization algorithms. Algorithm 2 depicts the Adam optimization process, which the proposed model uses.

---

**Algorithm 2:** Optimization Process

---

**Input**
Step length $\alpha$;
Erate ← $\beta$1, $\beta$2 ∈ [0, 1);
Rf ← $\mu(\tau)$;
Ip ← $\tau$0
**Begin**
    t equal of 0
    **While** t0 not meet
        t = t + 1;
        Calculatetime ← t: gt = $\nabla \tau \mu t(\tau t+1)$
        Udeviation ← $\lambda t = \beta 1 \cdot \lambda t - 1 + (1 - \beta 1)$
        renovate: $\tau t = \tau t - 1 - (\alpha/(1 - \beta t1) * \lambda t / u t$
    **Return** $\tau$t
    **Output** parameters $\tau$t
**End**

---

## 4. Results and Discussion

Table 7 displays the experimental dataset's specific parameters, including batch size, sentence length and Dropout value. The experiment's server hardware environment consists of a Windows 8.1 operating system, an HP Nvidia graphics card 7370 with 8 GB of memory, and a 1 TB hard drive.

**Table 7.** Experimental Parameter Setup.

| Parameters | Values | Parameter | Value |
|---|---|---|---|
| Batch-size | 128 | Sentence length | 70 |
| Convolution window-size | 3-4-5 | Word vector dimension | 300 |
| LSTM Hidden Number | 256 | Number of convolution kernels | 256 |
| Learning rate | 0.0001 | Activation function | SoftMax |

### 4.1. Evaluation Criteria

Accuracy, precision, recall, and F1- score are the indicators the proposed model uses to evaluate the outcomes of the classification of ED and SA samples. Respectively, accuracy is the term of the evaluation metric used to measure overall performance. The above-described evaluation criteria can be broken down into four distinct categories: True Positives (TP), True Negatives (TN), False Positives (FP), and False Negatives (FN).

The number of samples with positive emotions that were accurately identified as positive samples is denoted by the acronym TP. In contrast, the number of samples with negative emotions mistakenly predicted as positive samples is indicated by the term FP. According to TP, FP, TN is the percentage of negative-emotion samples that were correctly projected to be negative, whereas FN represents the number of positive-emotion samples that were incorrectly predicted to be negative. Furthermore, the performance matrix for the evaluation criteria Equation (36) calculates accuracy as the percentage of correctly predicted outcomes as follows:

$$\text{Accuracy} = \frac{\text{Tp} + \text{TN}}{\text{TP} + \text{TN} + \text{FN} + \text{FP}} * 100 \tag{36}$$

Equation (37) calculates precision as the percentage of text that was accurately classified into this category out of all text that was classified into this category.

$$\text{Precision} = \frac{\text{Tp}}{\text{TP} + \text{FP}} * 100 \tag{37}$$

Equation (38) calculates recall, which is the percentage of texts correctly assigned to this category across all texts in this category:

$$\text{Recall} = \frac{\text{Tp}}{\text{TP} + \text{FN}} * 100 \tag{38}$$

The F1-score is an indicator of the accuracy of a test, and it is derived from the precision and recall using Equation (39):

$$\text{F1} - \text{score} = 2 * \frac{\text{Precision} * \text{Recall}}{\text{Precision} + \text{Recall}} \tag{39}$$

### 4.2. Corpus Statistics

As mentioned earlier, we developed a sentiment analysis corpus and an emotion detection base corpus. Our emotion detection corpus has 1021 annotated sentences, while our sentiment analysis collection has 20,251 samples. In addition, Table 8 comprehensively summarises our original cleaned and tagged corpus in terms of sentence counts. Furthermore, it displays the combined frequency of the labelled corpora for emotion detection and sentiment analysis. After completing the annotation process, we set a different count of samples that belonged to each of the distinct emotions.

**Table 8.** Emotion Detection and Sentiment Analysis Dataset Description in terms of Sentence Count.

| Emotion Detection | | Categories Statistic | |
|---|---|---|---|
| **Emotion Sentences** | **1021** | **Emotions Number** | **6** |
| Total words in emotion | 6822 | Happy | 227 |
| | | Sad | 172 |
| Unique word count in emotion | 2149 | Anger | 213 |
| | | Love | 187 |
| Average words in a sentence | 18 | Fear | 173 |
| | | Neutral | 248 |

**Table 8.** *Cont.*

| Emotion Detection | | Categories Statistic | |
|---|---|---|---|
| **Emotion Sentences** | **1021** | **Emotions Number** | **6** |
| **Sentiment Analysis** | | **Categories statistic** | |
| Count of total sentences for sentiment analysis | 20,251 | Positive | 8928 |
| The average number of words in a sentence for sentiment analysis | 16 | Negative | 6016 |
| Unique word count in sentiment analysis | 42,684 | Neutral | 5290 |
| Total words | | 268,819 | |

*4.3. Model Results on Epoch*

A DL epoch approach is a full iteration that involves moving all of the training data in both the forward and the backward direction during the process. The experiment is built on the CNN-LSTM model. Every other parameter stays the same, with the exception of Tf-idf, count vectorization, and wrod2vec; the epoch parameters are the only ones that change. Figure 4 illustrates the training accuracy of the emotion detection algorithm across a number of epochs.

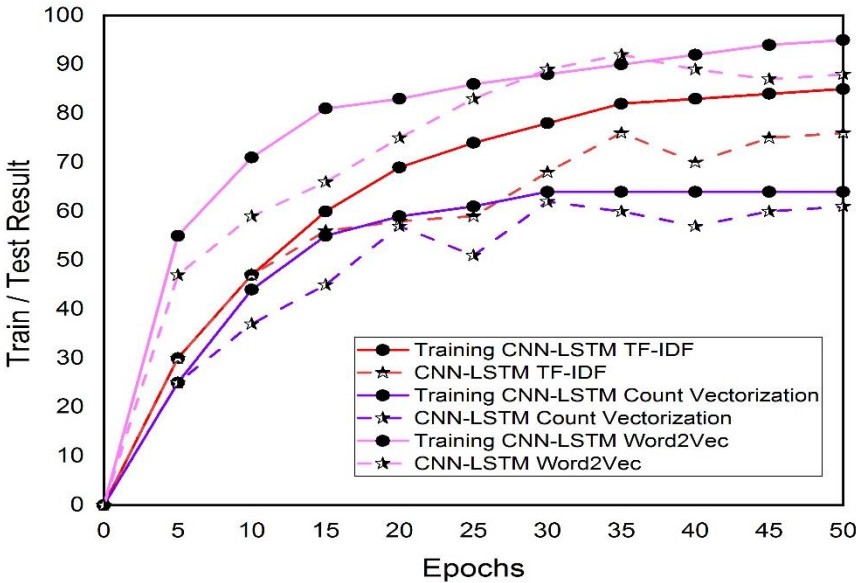

**Figure 4.** Comparison of Training and Testing Sets Results for Emotion Detection with Different Epochs.

As can be observed in Figure 4, as the epoch grows, the training set's accuracy in the text emotion detection task keeps getting better, while it decreases and subsequently increases for the test set shown by the dotted line in Figure 4. The test set accuracy peaks when the 35 epoch is trained for CNN-LSTM Word2Vec, then declines, presumably because of overfitting at the start of training. It can be stated that the outcomes of text emotion analysis are affected by either having too many or too few epoch values. The ideal results will not be reached if there are not enough epochs; however, if there are an excessive number of epochs, the model will end up overfitting the training data and will have a bad performance on the test set.

Figure 5 illustrates our second task: train and test our sentiment analysis dataset to obtain acceptable accuracy over several epochs. The training set's accuracy in the SA task improves over the epoch, similar to the prior aim, as seen in Figure 4, while it also declines

and then increases for the test set denoted by the dotted line in Figure 5. The test set accuracy peaks when the 50 epoch is trained for CNN-LSTM Word2Vec.

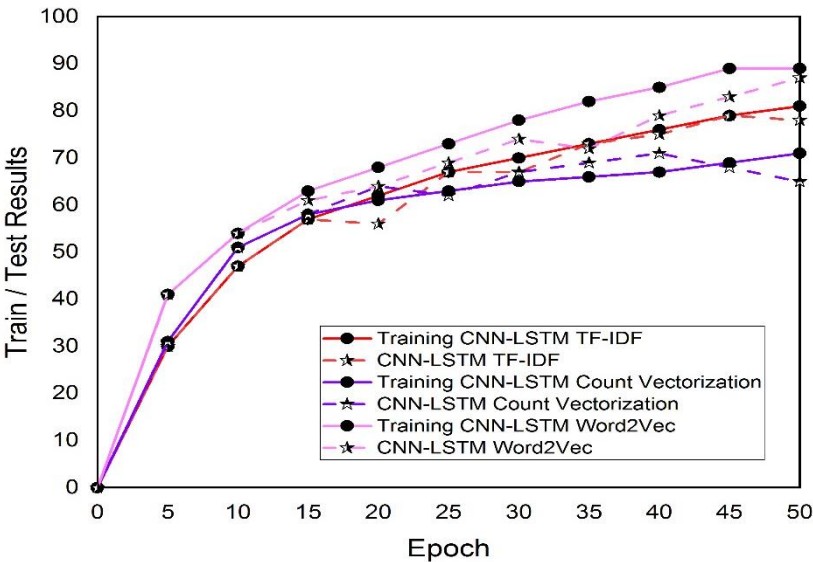

**Figure 5.** Training and Testing Comparison Results Sets for Emotion Detection with Different Epochs.

As a result, the value of epoch is crucial for assessing the model's effectiveness; based on the findings of the experiments, an epoch is set to 35 for emotion detection, and for sentiment analysis, it is set to 50. The model is now performing analysis in an optimal manner.

### 4.4. Experimental Results Dropout Value

Dropout involves the removal of some neurons and the subsequent updating of the weight and bias terms in the remaining neurons throughout forward and reverse propagation. After that, neurons are removed using a probability that has been previously determined, the weight and bias term is adjusted, and the procedure is repeated until the NN is properly trained. The dropout value is changed under the assumption that all other parameters remain the same. Figure 6 displays the accuracy outcomes of the proposed models for the emotional detection and sentiment analysis dataset.

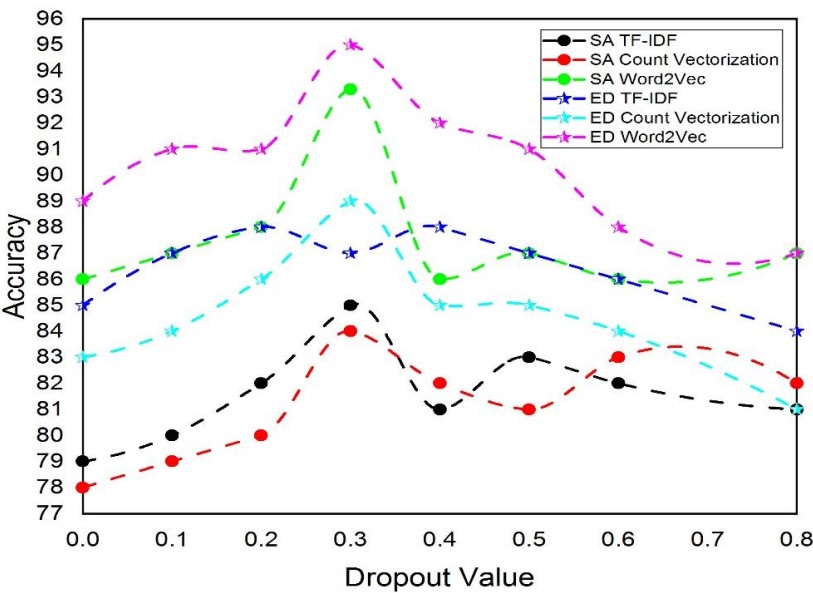

**Figure 6.** Comparison of Results with Dropout Values for Emotion Detection and Sentiment Analysis.

Figure 6 indicates that when employing the CNN-LSTM Word2Vec model, both the accuracy for emotion detection (95%) and the accuracy for sentiment analysis (93.3%) are at their highest (with a dropout value of 0.3). The accuracy performance is poor when the dropout number is either high or excessively low. This is because it is simple to slide into overfitting when training involves too many neurons and the dropout value is too low. When dropout is too high, too many neurons are left, which causes underfitting. As a result, in the subsequent comparative experiment, the dropout value of the model provided is set to 0.3. the dropout of each comparative experiment is adjusted several times, and the best experimental data are picked, as shown in Table 9.

**Table 9.** Achieved Accuracy using Dropout values.

| Emotions | Dropout Value | Algorithm | Techniques | Accuracy | Precision | Recall | F1-Score |
|---|---|---|---|---|---|---|---|
| Emotion Detection | 0.3 | CNN-LSTM | BOW | 89% | 90% | 88% | 89% |
| | 0.2 | | TF-IDF | 87% | 88% | 86% | 87% |
| | 0.3 | | Word2Vec | 95% | 94% | 96% | 95% |
| Sentiment Analysis | 0.5 | CNN-LSTM | BOW | 84% | 81% | 84% | 83% |
| | 0.6 | | TF-IDF | 85% | 81% | 86% | 84% |
| | 0.3 | | Word2Vec | 93.3% | 94% | 93% | 93% |

*4.5. Comparison of Training Results with Other Algorithms*

The proposed model integrates CNN and LSTM to improve each individual's strengths. Due to its ability to better consider past and future data, LSTM outperforms competing algorithms in emotion detection and sentiment analysis. Figure 7 shows the CNN-LSTM Word2Vec model's accuracy compared to CNN, LSTM, RNN, and ANN models. Experimental findings for ED and SA datasets show that the proposed model achieved higher accuracy than other models, about 95% for ED and 93.3% for SA, demonstrating its feasibility.

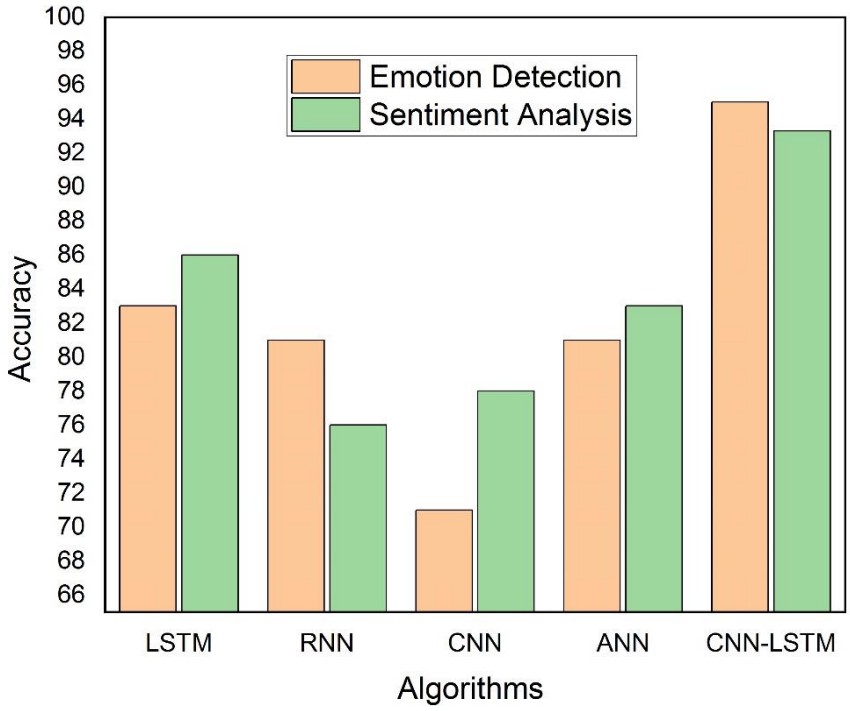

**Figure 7.** Proposed Method Results Compare some Machine Learning Fundamental Algorithms.

*4.6. Comparative Analysis*

The statistics of the experimental results are shown in Table 10. the suggested model can be compared and contrasted with the models in references [19,30] and [17] in order to show how well it performs. According to the table, the proposed model has the highest analysis performance, with an overall analysis accuracy of 95% in emotion detection and 93.3% in sentiment analysis.

**Table 10.** Comparative Analysis with Previous Work.

| Emotion Detection | | | | | |
|---|---|---|---|---|---|
| **Reference** | **Algorithm** | **Accuracy** | **Precision** | **Recall** | **F1-Score** |
| Ref [19] | Bi-LSTM | 85% | 84% | 87% | 85% |
| Ref [36] | Deep-EmoRU | 82% | 84% | 82% | 83% |
| Proposed | CNN-LSTM | 95% | 94% | 96% | 95% |
| Sentiment Analysis | | | | | |
| Ref [24] | KNN (IBK) | 67% | 68% | 67% | 67% |
| Ref [37] | Urdu Sentiment | 89% | 86% | 90% | 88% |
| Proposed | LSTM-CNN | 93.3% | 94% | 93% | 93% |

## 5. Conclusions

The network environment is becoming more complex as the volume of information increases. Research into emotion detection and sentiment analysis has been a priority in the field of natural language processing because of its relevance to the understanding of public opinion. As a result, at this point, accurate emotion detection and sentiment analysis have the significant scientific value given the ongoing advancement of artificial intelligence. Aiming the problem's emotion detection and sentiment analysis in English and Chinese has received a lot of attention in the last decade, but poor-resource languages such as Urdu have been mostly disregarded, which is the primary focus of this research. Additionally, due to the lack of a publicly accessible corpus, most low-level language problems, such as existing deep learning methods in ED and SA have poor analytical accuracy. To develop the CNN-LSTM model, the proposed model combines LSTM and CNN. This is because the LSTM hidden layer depends on the outcomes of the previous period, whereas CNN obtains deep features. Thus, the accuracy is improved based on the corpus collected for emotion detection and sentiment. As a result, the accuracy is enhanced using the corpus collected for sentiment and emotion detection. In the future, we will propose replacing the LSTM with a bidirectional BILSTM network to improve the model's analytical efficiency.

**Author Contributions:** Conceptualization, F.U., X.C., S.B.H.S. and M.A.H.; methodology, F.U. and M.A.H.; software, F.U., S.M. and M.A.H.; validation, F.U., N.S. and M.A.H.; formal analysis, F.U., X.C. and M.A.H.; investigation, F.U., N.S. and M.A.H.; resources, F.U. and M.A.H.; data curation, F.U., N.S., S.M. and M.A.H.; writing—original draft preparation, F.U., S.B.H.S. and M.A.H.; writing—review and editing, F.U., S.B.H.S. and M.A.H.; visualization, F.U. and M.A.H.; supervision, X.C., N.S. and M.A.H.; project administration, F.U., X.C., S.M. and M.A.H.; funding acquisition, S.B.H.S. and S.M. All authors have read and agreed to the published version of the manuscript.

**Funding:** This Study was funded by the Vice presidency for Graduate Studies, Business and Scientific Research (GBR) at Dar Al Hekma University, Jeddah Saudi Arabia. The authors extend their sincere gratitude and thank to Dar Al Hekma University for its support.

**Data Availability Statement:** All Numerical and graph is available in the manuscript and dataset will be provided on request.

**Conflicts of Interest:** The authors declare no conflict of interest.

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
