# Peer review of "A Novel Approach for Emotion Detection and Sentiment Analysis for Low Resource Urdu Language Based on CNN-LSTM"

_electronics, doi:10.3390/electronics11244096_

Round 1
Reviewer 1 Report
The authors propose an architecture for Sentiment and Emotion detection in the Urdu language. The proposed architecture combines CNN and LSTM for the classification task and uses word2vec as embedding.
The authors should better justify why they consider CNN with LTSM. CNN with LSTM has been used multiple times before, including for the SA task. Thus, I do not see any novelty in this.
Article [1] uses a CNN-LSTM architecture too on English and Urdu languages. The solution from [1] uses different word embeddings (Word2Vec, Glove, Fasttext) and not just word2vec (as in this article).
Related work is lacking. The authors should discuss other SA/ED solutions that employ CNN&LSTM.
Besides [1], article [2] presents experiments with different architectures that combine CNN with multiple layers of Bi(LSTM) or Bi(GRU). As [1], they also use as embeddings Word2vec, Fasttext, and Glove. This should be something to read, as the authors proposed CNN with BiLSTM as future work.
Related to the previous paragraph, another discussion in Related work can describe SA solutions that use contextualized embeddings and not just static embeddings. An example of contextualized embedding is the proposed solution from article [2].
The authors should clarify what is the meaning of "Result" in Tables 1 and 2. What metric is used: accuracy, precision, recall? Modify the name of the Result column with the name of that metric. If the solutions use different metrics then they can not be compared like that. On the same topic, Figure 7 should have 'Accuracy' and not 'Results' (on the 0y axis).
In the results, the authors compare their solution with simple CNN, LSTM, RNN (which makes no sense as vanilla RNN is always worst than the other types of RNN - LSTM or GRU), or ANN.
The authors should remove citations 31 to 38, as they are only templates and not actual citations.
[1] L. Khan et al. "Deep Sentiment Analysis Using CNN-LSTM Architecture of English and Roman Urdu Text Shared in Social Media"
[2] C Truica et al. "Topic-Based Document-Level Sentiment Analysis Using Contextual Cues"
Author Response
Many Thanks for your Valuable Suggestions. Please find the overall review file attached to this review page.

Reviewer 2 Report
In this study, we create a corpus of 1021 sentences for emotion detection and 20251 sentences for sentiment analysis, both obtained from various areas, and annotate it with the aid of human annotators from six and three classes, respectively. It is a well-structured paper with interesting results. However, it requires further improvements.
(1)The abstract should be improved. Your point is your own work that should be further highlighted.
(2)The parameters in expressions are given and explained.
(3) The method in the context of the proposed work should be written in detail
(4) The values of parameters could be a complicated problem itself, how the authors give the values of parameters in the used methods.
(5) The literature review is poor in this paper. You must review all significant similar works that have been done. For example, https://doi.org/10.1109/JSTARS.2021.3059451; https://doi.org/10.1016/j.ins.2022.11.019; https://doi.org/10.3934/mbe.2023090; https://doi.org/10.1016/j.ymssp.2022.109422 and so on.
(5) In stection 3.7 of Optimization Algorithm, which agorithm is used by the authors?
(6) At Line 364, how to determine the Parameter Setting in Table 7?
Author Response
Many thanks for your valuable Comments. Please find the Review file herewith this review response window.

Reviewer 3 Report
Very good and intersting work
Author Response
Many Thanks for your Valuable Comments. Please find the review herewith this reviewer's response page.

Reviewer 4 Report
1. We know that sometimes there are multiple emotions in a sentence, such as sadness, anger and disgust. How to deal with complex and multiple emotions?
2. Some pictures are not clear, especially the text in the pictures is too small and unclear, such as figure 3.
3. The variables or parameters in the equations are not specified.
4. It is recommended that algorithms 1 and 2 be expressed using tables.
5. This is an error on line 428 in page 15. Figure 3 should be figure 5, moreover the title of figure 5 is wrong.
6. Text legends in figure 4~6 should be kept within the border of the figures.
7. English grammar and expression need to be carefully checked and improved by the authors.
Author Response
Many Thanks for your Valuable Comments. Please find the response file herewith attached to this reviewer response page.

Round 2
Reviewer 2 Report
My comments in the first round of review have all been well answered, and I think the paper can be accepted.